# The Role of MYC and PP2A in the Initiation and Progression of Myeloid Leukemias

**DOI:** 10.3390/cells9030544

**Published:** 2020-02-26

**Authors:** Raffaella Pippa, Maria D. Odero

**Affiliations:** 1Medical Oncology Department, Sidney Kimmel Cancer Center, Thomas Jefferson University, Philadelphia, PA 19107, USA; 2Centro de Investigación Médica Aplicada (CIMA), University of Navarra, 31008 Pamplona, Spain; 3Biochemistry and Genetics Department, University of Navarra, 31008 Pamplona, Spain; 4IdiSNA, Instituto de Investigación Sanitaria de Navarra, 31008 Pamplona, Spain

**Keywords:** PP2A, MYC, myeloid leukemia, AML, CML

## Abstract

The MYC transcription factor is one of the best characterized PP2A substrates. Deregulation of the MYC oncogene, along with inactivation of PP2A, are two frequent events in cancer. Both proteins are essential regulators of cell proliferation, apoptosis, and differentiation, and they, directly and indirectly, regulate each other’s activity. Studies in cancer suggest that targeting the MYC/PP2A network is an achievable strategy for the clinic. Here, we focus on and discuss the role of MYC and PP2A in myeloid leukemias.

## 1. Introduction

Myeloid cells, including erythrocytes, monocytes/macrophages, granulocytes, and megakaryocyte/platelets and dendritic cells, are key effector cells of the innate immune defense against invading micro-organisms [1]. These cells are continuously generated from hematopoietic stem cells in the bone marrow through a well-timed and regulated process named myelopoiesis [2]. Deregulation of proliferation and lineage differentiation due to the genetic and epigenetic aberrations is linked to blood cell disorders such as acute myeloid leukemia (AML) and chronic myeloid leukemia (CML) [3].

Research in hematology has fundamentally improved our understanding of the biology of cancer and resulted in many innovative discoveries [4]. Many genetic aberrations have prognostic impact on myeloid leukemias, including the overexpression of the MYC, CIP2A, and SET oncogenes. MYC is a master transcription factor that regulates a wide spectrum of target genes, and is found altered in several types of cancer [5,6]. CIP2A and SET are endogenous inhibitors of the phosphatase 2A (PP2A), a tumor suppressor protein that is recurrently inactivated in acute and chronic myeloid leukemias [7]. PP2A is a key regulator of many oncoprotein signaling pathways involved in the regulation of cell growth and survival, including MYC. Mainly, PP2A activity leads to MYC degradation [8]; therefore, its role in regulating MYC biology is fundamental to maintain a correct balance between normal and aberrant phenotype. In this review, we focus on the role of MYC and PP2A in the myeloid leukemia initiation and progression.

### 1.1. Acute Myeloid Leukemia (AML)

AML is a clonal malignancy that results from genetic and epigenetic abnormalities which leads to uncontrolled proliferation and impaired differentiation of hematopoietic stem and progenitor cells [9,10,11]. Cytogenetic analysis for the detection of chromosomal abnormalities was the first prognostication tool in AML and still remains the backbone of current AML genomic classification [12]. With the advancement of genetic and next-generation sequencing approaches, recurrent mutations in AML have also been characterized [13,14]. The genetic profile of AML is notably heterogeneous, and only a few mutations (i.e., FLT3, NPM1, and DNMT3A) are present in more than a quarter of AML patients [15]. Transcription factors are a class of genes that are frequently altered in AML, such as PML, EVI1, GATA2, RUNX1, ETV6, and C/EBPα, which establish and maintain genetic networks governing the genesis and function of blood stem and progenitor cells [16]. Furthermore, a recent genomic analysis in over 1500 AML patients identified novel hotspot mutations in another critical transcription factor, such as MYC [17]. MYC high expression is also common in AML.

Scientific understanding of the AML biology has led to the development of targeted treatment approaches [4], i.e., the all-trans retinoic acid (ATRA) treatment for acute promyelocytic leukemia (APL) that carries the PML-RARα translocation [18] or the use of FLT3 kinase inhibitors in FLT3-mutated cases [19,20]. Nevertheless, the treatment of AML has been mostly unmodified for the past decades. Although many patients with AML respond to induction chemotherapy, relapse is common and represents the primary cause of treatment failure [9]. Interestingly, bone marrow environment has been suggested to play a role in this unresponsive phenotype, by providing shelter to leukemic stem cells, whose persistence eventually leads to therapy resistance [21].

### 1.2. Chronic Myeloid Leukemia (CML)

CML is a type of leukemia with an initial chronic phase (CP) characterized by a massive expansion of myeloid precursors and mature cells, followed by a late acute phase termed “blast crisis” (BC), which resembles AML [22]. CML biology is characterized by the presence of the reciprocal translocation t(9;22)(q34;q11) that generates a BCR-ABL fusion protein with aberrant and uncontrolled activity [23]. The development of a BCR/ABL kinase inhibitor, imatinib, was a significant major improvement in CML treatment [24]. However, imatinib, and second and third-generation tyrosine kinase inhibitors’ effectiveness is often inadequate as resistant CML stem cells and residual disease persist in many patients [25,26,27]. Genomic sequencing of CML patients has identified additional genetic changes, like mutations of the tumor suppressor genes RB1, TP53, and CDKN2A [28], and highlighted the fact that BCR-ABL kinase mutations are only identified in approximately 50% of patients with poor responses and disease progression [29]. Therefore, progress in our understanding of CML biology is still required.

### 1.3. MYC

MYC is a master transcription factor of the helix-loop-helix leucine zipper (HLHZ) family [30]. MYC forms an heterodimer with MAX and binds to E-box sequences in the regulatory regions of numerous target genes [31] involved in the control of proliferation and differentiation, metabolism, cell growth, among other processes [32,33]. The expression of MYC is tightly controlled in normal cells. However, the aberrant activity of MYC is one of the most common events in solid and hematopoietic neoplasias [31].

The MYC gene was first discovered as the oncogene carried by a retrovirus that induced a myeloid neoplasm in chicken, i.e., myelocytomatosis [34] and a few years later identified in human Burkitt lymphoma and in mouse plasmacytomas [35,36]. MYC is amplified or deregulated in both AML and CML; however, the mechanisms that lead to MYC overexpression in AML are still not well-defined. It has been suggested that the aberrant activity of rearranged transcription factors (i.e., RUNX1-ETO, PML-RARα, and PLZF-RARα) might increase MYC expression. Trisomy 8, where MYC is localized, could represent another potential mechanism. MYC upregulation is usually moderate in myeloid leukemias, yet, enough to generate radical changes in the myeloid precursor differentiation [37].

MYC is a short half-live protein; thus, its post-transcriptional regulation plays an essential role in its stability and function. Two interdependent phosphorylation sites that are critical for the regulation of MYC have been identified. Activation of the ERK cascade or cyclin-dependent kinases (CDKs) leads to the phosphorylation of MYC at Serine 62 (S62) [38]. This modification enhances MYC DNA binding and gene regulation. Phosphorylation of S62 also primes MYC for glycogen synthase kinase 3 (GSK3)-mediated phosphorylation at Threonine 58 (T58) [39], which initiates MYC turnover [8]. Dual phosphorylated MYC (S62 and T58) becomes target of the PP2A, which dephosphorylates the stabilizing S62 residue and marks MYC for ubiquitin-mediated proteosomal degradation [40] (Figure 1).

### 1.4. MYC in Myeloid Leukemias

The mechanisms behind MYC-driven oncogenesis in AML are still underexplored. Different murine models, either using engraftment of hematopoietic precursors over-expressing MYC or transgenic mice, demonstrated the myeloid leukemogenic activity of MYC [37] and highlighted the significance of MYC-driven abrogation of apoptosis as key molecular mechanism [41,42]. In APL, MYC cooperates with PML-RARα to accelerate the development of leukemia [43], while in AML, MYC protein expression is a poor prognostic factor, particularly in patients with high risk of relapse [44,45]. It has been reported that the inhibition of MYC in AML cell lines drastically induces cell differentiation [46], and studies in mice have demonstrated that even transient inactivation of MYC yields tumor regression [47], supporting the hypothesis that targeting MYC regulation could be a highly pursued goal to treat cancer patients.

In CML, it has been widely reported that the BCR-ABL aberrant kinase activity induces MYC transcription [48] and translation, and that the progress into blast crisis depends on this BCR-ABL-induced MYC expression [49]. The generation of a mouse model with an inducible BCR-ABL along with sequentially and spontaneously activated mutations, helped to recapitulate the human CML biology. Interestingly, Giotopoulos et al. identified potentially targetable pathways in CML such as ERG, MEK, RAF, JAK1/2, and confirmed that MYC is crucial for CML-progression to BC [50]. Additionally, MYC and its partner MAX bind to the BCR promoter, enhancing BCR-ABL mRNA and protein content, creating a feedback loop that is key for CML transition to BC [51]. Nevertheless, Reavie et al. recently showed that MYC is essential but not sufficient for CML induction and progression. Additional mutations, such as p53 loss, are needed to sustain CML pathobiology [52].

Yet, high levels of MYC correlate with poor response to imatinib and, at molecular level, MYC induces genomic instability [53]. Also, MYC’s negative regulation of p27^kip1^ protein stability seems to be critical. By inducing SKP2, a component of the ubiquitin ligase complex that targets p27^kip1^ for degradation, MYC prevents cell differentiation [54].

### 1.5. PP2A

Reversible phosphorylation is one of the mechanisms that cells use to maintain normal homeostasis [7], yet the role of phosphatases in cancer has just started to be fully considered. PP2A is a serine-threonine phosphatase that dephosphorylates serines or threonines residues of key factors involved in the regulation of cell proliferation, survival, and differentiation and that together with PP1, contributes >90% of the phosphatase activity of the cell [55]. Such wide-ranging substrate specificity is achieved by the formation of at least 100 distinct heterotrimeric holoenzymes, each containing a common core enzyme formed by the catalytic (C) and scaffold (A) subunits and a third variable regulatory subunit (B) that determine substrate specificity and subcellular localization [56]. In humans, the C and A subunits are each encoded by two different genes, giving rise to two isoforms of the C subunit (PPP2CA and PPP2CB), and A subunit (PPP2R1A and PPP2R1B). For the B-type subunits, 15 human genes have been described, giving rise to more than 23 isoforms belonging to four different families, PR55 ⁄ B (or B55, encoded by PPP2R2A to PPP2R2D), PR61 ⁄ B’ (or B56, encoded by PPP2R5A to PPP2R5E), PR72 ⁄ B’’ (encoded by PPP2R3A to PPP2R3C), and PR93 ⁄ PR110 ⁄ B’’’ (or the striatins, encoded by STRN, STRN3 and STRN4) (Table 1). This diversity in composition creates specificity and forms the basis for PP2A’s multiple physiological functions [57,58]. Particularly, complexes containing PP2A B56α, B56γ, B55α, and PR72/PR130 have been reported to specifically activate MYC, WNT, ERK MAPK, and PI3K/AKT pathways and, thus, considered actionable targets [56,59,60,61].

PP2A is frequently inactivated in both AML and CML [64,68], and several studies now recognize PP2A inactivation as an essential requirement for transforming human cells [56]. Mutations, mostly in the A subunit that impair the phosphatase function or assembly, have been reported in solid tumors. However, their incidence in myeloid leukemia is low, except for the PP2A-C subunit haploinsufficiency in AML with deletions of chromosome 5, a region that includes other tumor suppressor genes [65]. Altered expression of PP2A subunits is a recurrent event in AML. Mainly, PP2A-B55α is under-expressed in primary blast cells and correlates with increased AKT phosphorylation and shorter complete remission [7,69]. At protein level, B55α content also associates with increased MYC activity [70]. Nevertheless, current evidence suggests that the inhibition of PP2A is mostly achieved through non-genomic mechanisms, such as post-translational modifications of its subunits and/or overexpression of PP2A endogenous inhibitors [7].

The activity and localization of PP2A can be regulated by post-translational modifications [73]. Methylation and phosphorylation of residues from PP2A-C subunits modulate the formation of the complex. For instance, methylation of leucine 309 (L309) in the catalytic PP2A-C by the leucine carboxyl methyltransferase 1 (LCMT1) [74] is required for binding of the PR55/B55 subunit [75]. Phosphorylation of tyrosine 307 (Y307) considerably impairs PP2A phosphatase activity by inhibiting the interaction of PP2A-C with the PTPA-activating protein, which enables the assembly of the holoenzyme [58,76], and the regulatory subunits PR55/B55 and PR61/B56 [77]. Actually, increased phosphorylation of Y307 of PP2A-C inversely correlates with disease prognosis in AML and CML [68,78]. In contrast, threonine 304 (T304) phosphorylation prevents the assembly of PR55/B55 to the core enzyme [75].

In AML, our group and others showed that the hyperphosphorylation of PP2A-C and consequent inactivation is observed in 78% of AML patients, along with high levels of PP2A endogenous inhibitors SET and CIP2A [68,79,80,81]. In c-KIT-mutated AML, oncogenic receptor tyrosine kinase c-KIT, which activates proliferation, differentiation, and survival signaling pathways, requires PP2A inactivation for leukemogenesis. Roberts et al. have shown that myeloid precursor cells expressing oncogenic mutant c-KIT receptors display significantly reduced PP2A activity compared to c-KIT negative or wild type c-KIT. Furthermore, inhibition of PP2A by mutant c-KIT is associated with reduced protein expression of PP2A subunits, together with altered expression of PP2A endogenous inhibitor SET oncogene [63]. PP2A inhibitors are described in more detail in the next paragraph.

In CML, functional inactivation of PP2A by BCR-ABL is essential for the progression into blast crisis. One suggested mechanism of inactivation is the increased expression of the PP2A-Aα subunit observed in BCR-ABL+ cells, as the Aα subunit could sequester the catalytic or regulatory subunits and act as a PP2A inhibitor [82]. However, it is generally believed that BCR-ABL enhances the expression of the endogenous PP2A inhibitor SET with subsequent loss of PP2A activity [64,83]. Imatinib-derived inhibition of BCR-ABL dramatically reduces SET expression and restores PP2A activity, resulting in the dephosphorylation of STAT5, ERK1/2, AKT, BAD, and JAK2, substrates of both BCR-ABL and PP2A [78,84].

### 1.6. PP2A Endogenous Inhibitors

Recent reports have highlighted the role of the PP2A endogenous inhibitors ANP32A, SET, CIP2A, and ARPP19 (Figure 2), and their role in cancer progression [85,86]. Particularly, their alterations in myeloid leukemia are reported in Table 2.

ANP32A, (acidic nuclear phosphoprotein 32a), also known as I1PP2A (Inhibitor 1 of PP2A), is a potent inhibitor of PP2A catalytic subunit [87,88]. It is overexpressed in primary AML cells [89] and is a critical factor that contributes to acute megakaryoblastic leukemia progression [101].

SET, also known as Inhibitor 2 of PP2A (I2PP2A) or TAF-1 or PHAPII, is a multitask protein [102] that along with being a histone chaperone [103] and inhibitor of histone acetylation [104] binds directly to the catalytic subunit C of PP2A, impairing its activity [90,91]. This oncoprotein is overexpressed in both solid and hematological tumors [64,79]. In AML, SET is highly dependent on MYC-transcriptional activity and recruitment of RUNX1 and GATA2 on its promoter [102]. By impairing SET binding to PP2A, with molecules like FTY720 [82,105], CM-1231 [106], or OP449 [107,108] it is possible to re-establish PP2A activity, inhibiting tumor growth [91,109] and overcoming therapeutic resistance in preclinical models [92,108]. SET is known to interact with SETBP1 (SET-binding protein 1), which protects SET from protease cleavage. SETBP1 is also overexpressed in over 27% AML patients, and its high expression significantly correlates with shorter overall survival, mostly in patients over 60 years [79]. Mutations in the *SETBP1* gene have been found, and the coexistence of SETBP1 and ASXL1 mutations is known to promote leukemogenesis by repressing TGFβ pathway genes [110]. Genetic mutations in the SETBP1 gene have also been found in CML and other myeloproliferative disorders with no BCR-ABL translocation [111,112]. Along with its SET stabilizing function, SETBP1 additionally promotes self-renewal of myeloid progenitors in vivo, wherein BCR-ABL positive conditions help generating aggressive leukemias in recipient mice [113]. In addition, our group has recently shown that p38β potentiates the inactivation of PP2A mediated by SET by two mechanisms: facilitating cytoplasmic translocation of SET through phosphorylation of CK2, and directly binding and stabilizing the SET protein [114].

CIP2A, also named KIAA1524, is a scaffold protein that stabilizes MYC by inhibiting the PP2A-regulated de-phosphorylation on the residue S62 [93]. Structure analysis of the CIP2A-PP2A interaction revealed that CIP2A forms homodimers. The dimer is stabilized by its interaction with the PP2A-B56α and B56γ subunits [94]. CIP2A specifically interacts with PP2A-B56α holoenzyme, which is the most ubiquitously expressed isoform of the B56 family [115] and the one that associates with MYC to negatively regulate its protein levels and activity [38]. Thus, in the absence of PP2A-B56α-mediated control, MYC can be aberrantly expressed. By preventing PP2A-B56α interaction with MYC, CIP2A gained scientific attention, and it is now considered a promising target for tumor treatment, particularly in AML [81,96,116], where it is overexpressed and promotes cell growth and neoplastic transformation [81]. Additionally, CIP2A has been reported to be a translocation partner with the mixed-lineage leukemia (MLL) gene [117]. In CML, CIP2A is also biologically and clinically important [96,118], as high levels correlate not only with increased levels of MYC, but with upregulation of the antiapoptotic protein BCL-XL [97], another important target of PP2A [119]. Additionally, CIP2A mRNA and protein content is regulated by MYC, creating a positive feedback loop between the two oncogenes in cancer [95]. By inhibiting MYC, by treating the cells with an inhibitor of MYC interaction with its partner MAX through the basic-helix-loop-helix-leucine zipper domain, 10058-F4 [120], Lucas et al. demonstrated that the levels of CIP2A and SET could be dramatically reduced, confirming once again that studying the inhibition of MYC is worth it [121].

Among the other PP2A negative regulators, ARPP19, together with ENSA and BOD1, associates with and inhibits PP2A to promote mitotic entry [98,122,123]. While its role in tumor progression is still underexplored, a recent study revealed that ARPP19 mRNA expression is an independent predictor for relapse in AML, and it might promote cell survival by regulating CIP2A and MYC expression [100].

### 1.7. Targeting the MYC/PP2A Axis in Leukemia

MYC is a desirable therapeutic target for several types of cancer. Yet, because of the absence of a clear ligand-binding domain and its unstructured nature, direct targeting of MYC is still a challenge [124,125].

Several molecules have been described as MYC inhibitors, although none of them are used in the clinic. The 10058-F4, which prevents the interaction of MYC with MAX, used in several preclinical studies, was able to reduce BCR-ABL kinase activity and CIP2A expression in CML patients [121]. In AML cells, it was reported that 10058-F4 inhibits growth, induces cell cycle arrest, and differentiation [126]. Besides, our group described that 10058-F4 treatment also reduces SET mRNA transcription, leading to PP2A re-establishment in AML cells [102].

Another indirect inhibitor of MYC is JQ1, a powerful inhibitor of BET bromodomain protein BRD4 which regulates MYC transcription [127]. The treatment with JQ1 or other BET inhibitors triggers MYC downregulation with consequent cell cycle arrest and apoptosis in mouse and human leukemia cells [128,129,130,131]. Some BET inhibitors are currently in early phase clinical trials for treating hematopoietic malignancies and solid tumors with exciting results [132]. However, toxic side effects [133] and cases of resistance have been reported [134].

Therefore, pointing at PP2A activation to circumvent MYC undruggability and target MYC protein stability is considered today a suitable and attractive approach. Even though the levels of MYC after PP2A activation have not been clearly defined in myeloid leukemias, it is known that the levels of PP2A activity indirectly correlate with the ones of MYC.

PP2A-activating drugs are known to be highly effective in reducing MYC activity in several types of cancers [135,136,137]. Notably, small molecules that prevent SET-PP2A interaction such as FTY720, OP449, and CM-1231 re-activates PP2A, inhibiting cell proliferation and promoting apoptosis in AML and CML cell lines and primary patient samples [62,63,68,83,92,106,108,138,139,140]. Combination of FTY720 or OP449 along with Idarubicin and Ara-C, drugs used in standard induction therapy in AML, or FLT3 inhibitors, as well as with tyrosine kinase inhibitors in CML, have a synergic effect and significantly reduced growth of leukemic cells [92,108,138,141].

## 2. Conclusions

Deregulation of the MYC oncogene, along with the inactivation of PP2A, is persistently found in myeloid leukemias, inducing tumor progression and conferring poor prognosis. Both proteins play pivotal roles in the initiation and development of the disease and constitute attractive therapeutic targets. As for other transcription factors, the development of molecules inhibiting MYC activity continues to be challenging. Thus, indirect approaches have been suggested. The discovery of the MYC/PP2A axis has set the foundation for the use of PP2A activating molecules to impair MYC stability. Ultimately, more knowledge of the MYC/PP2A biology is needed. Nonetheless, studies in leukemia cells and preclinical models hint that therapies targeting MYC/PP2A inhibitory network are a clinically feasible strategy for leukemia treatment.

## Figures and Tables

**Figure 1 cells-09-00544-f001:**
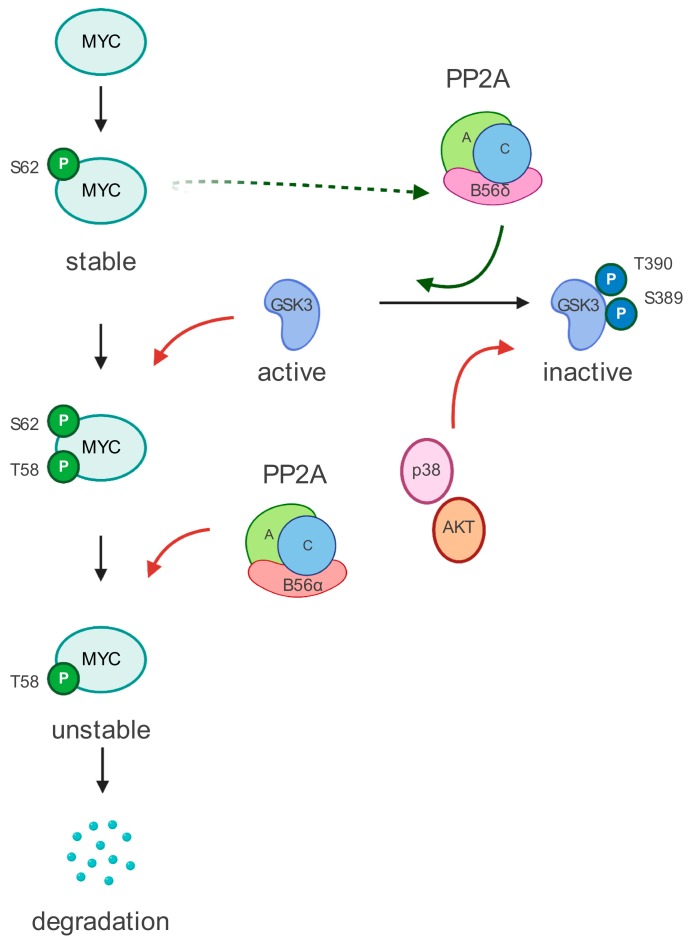
MYC regulation by PP2A. MYC undergoes two critical phospho-modifications on S62 and T58 residues. The phosphorylation on T58 is regulated by GSK3β. To be active, GSK3β has to be dephosphorylated. The PP2A complex, which includes the B56α subunit, dephosphorylates and activates GSK3β. Interestingly, the B56α subunit is transcriptionally activated by MYC. Moreover, the presence of a phosphorylated T58 residue also recruits the PP2A-B56α complex, which dephosphorylated the S62 phospho-residue. Dephosphorylation of S62 residue eventually targets MYC for ubiquitin-mediated proteosomal degradation. See text for more details.

**Figure 2 cells-09-00544-f002:**
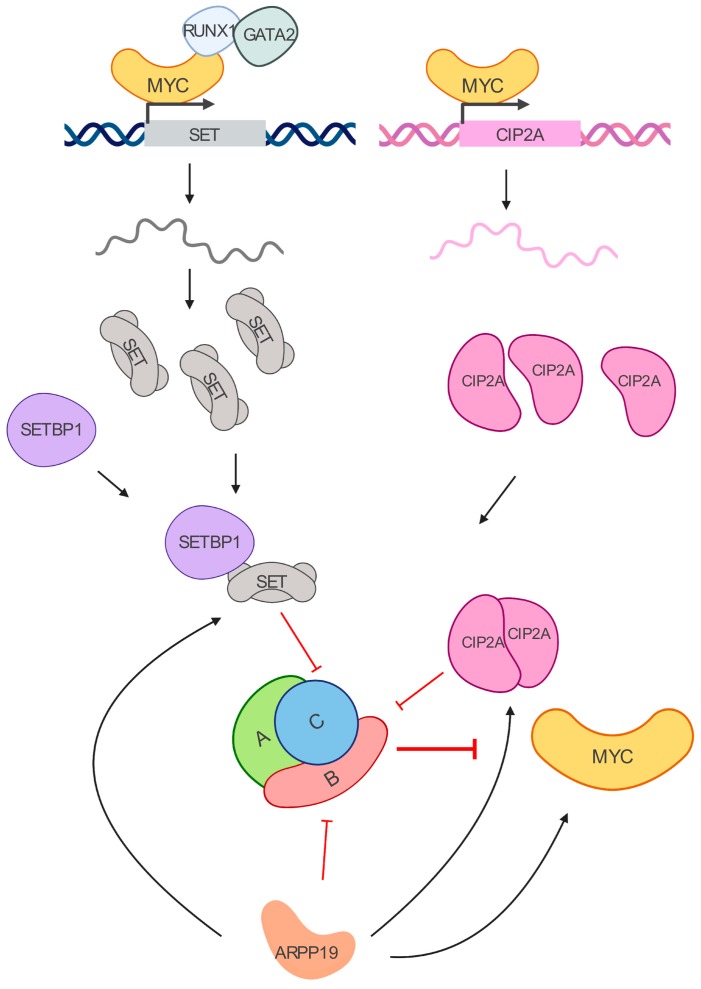
MYC/PP2A inhibitory network. MYC transcriptionally regulates SET and CIP2A. SET binds to the PP2A-C subunit, while CIP2A interacts with PP2A-B56𝛼 subunit. Their increased expressions lead to PP2A inhibition in leukemia cells. SETBP1 is a SET binding protein overexpressed in myeloid leukemias that prevents SET cleavage. ARPP19 is another inhibitor of PP2A that has been recently reported to enhance the levels of CIP2A, SET, and MYC in AML.

**Table 1 cells-09-00544-t001:** PP2A subunits and reported alterations in acute myeloid leukemia (AML) and chronic myeloid leukemia (CML).

Subunit.	Gene	Locus	Protein	Alteration in AML	Alteration in CML
A Structural	PPP2R1A	19q13.41	PR65/A𝛼	Downregulation [62] .Oncogenic c-KIT mutations decrease protein levels [63].	Increased levels in BCR/ABL cells [64].
	PPP2R1B	11q23.1	PR65/A𝛽	Downregulation [65]	
C Catalytic	PPP2CA	5q31.1	PP2A-C/C 𝛼	Downregulation in TP53 mutant AML cases [65]. Decreased expression in del5q AML [66]. Gene deletion in pediatric AML [67]. Hyperphosphorylation of Y307 [68].	Hyperphosphorylation of Y307 and T304 [64].
	PPP2CB	8p12	PP2A-C/C 𝛽		
B Regulatory	PPP2R2A	8p21.2	PR55/B55 𝛼	Oncogenic c-KIT mutations decrease protein levels [63]. Downregulation at mRNA and protein level [62,69,70].	
	PPP2R2B	5q32	PR55/B55 𝛽	High expression [65].	
	PPP2R2C	4p16.1	PR55/B55 𝛾	Downregulation [65].	
	PPP2R2D	10q26.3	PR55/B55 𝛿		
B’ Regulatory	PPP2R5A	1q32.3	PR56/B56 𝛼	Oncogenic c-KIT mutations decrease protein levels [63]. Gene amplification in pediatric AML [67].	
	PPP2R5B	11q13.1	PR56/B56 𝛽	Downregulation [68]. High expression [65]. Gene deletion in pediatric AML [67].	
	PPP2R5C	14q32.31	PR56/B56 𝛾	Downregulation [68]. Oncogenic c-KIT mutations decrease protein levels [63]. Gene deletion in pediatric AML [67].	Decreased expression in CML-BC compared to de novo CML [71].
	PPP2R5D	6p21.1	PR55/B55 𝛿	Oncogenic c-KIT mutations decrease protein levels [63]	
	PPP2R5E	14q32.2	PR55/B55 𝜀	Downregulation [72].	
B” Regulatory	PPP2R3A	3q22.2	PR72/PR130 or B” 𝛼		
	PPP2R3B	Yp11.32; Xp22.33	PR48/PR70 or B” 𝛽	Downregulation [65].	
	PPP2R3C	14q13.2	B” 𝛾or G5PR		
B’’’ Regulatory	STRN	2p22.2	Striatin		
	STRN3	14q13-q12	Striatin3		
	STRN4	19q13.2	Striatin4		

**Table 2 cells-09-00544-t002:** PP2A endogenous inhibitors and reported alterations in AML and CML.

Inhibitor	Mechanism	MYC Association	Alteration in AML	Alteration in CML
**ANP32A**	Binds to and inactivates PP2A-C [87,88].		High expression in primary AML cells [89].	
**SET**	Binds to and inactivates PP2A-C [90].	MYC regulates SET transcriptional expression [91].	High expression [68,80].	High expression [64].Associate with monosomy 7 and predict worse overall survival and progression-free survival [92].
**CIP2A**	Binds to PP2A A, B56 subunits, preventing the dephosphorylation of MYC [93,94]	CIP2A prevents PP2A-dependent dephosphorylation of MYC on S62 [93].MYC regulates CIP2A expression [95].	High expression [81,96].	High expression [97].
**ARPP19**	Binds and inhibits B55αδ in mitosis [98,99]	ARPP19 promotes MYC expression [100]	High expression [100]

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
