# Peer review of "The Role of MYC and PP2A in the Initiation and Progression of Myeloid Leukemias"

_cells, 2020, doi:10.3390/cells9030544_

Round 1

Reviewer 1 Report

This is a nice and comprehensive review on the roles of PP2A in myeloid leukemia. It is well written and the laboratory of the authors have contributed to this filed with relevant information. The review also discuss the state of the art so as to the therapeutic opportunities of therapeutic intervention in myeloid leukemia. The two tables and figures are very helpful and informative

I have some comments on minor concerns:

Line 82: Maybe it is advisable to cite here the discovery of MYC as a human oncogene reported 3 tears later (upon translocation in Burkittt lymphoma and mouse plasmacytomas, by Dalla Favera and Calame)

Lines 110-122: Myc and CML: They could consider to  cite here the first report on the association of MYC overexpression and CML (Albajar el al, Mol Cancer Res 2011)

Lines 161 and 189: AML is a complex of several entities with important differences so as to cell of origin, prognosis and frequency. Can the authors provide percentages of PP2A and SET1 deregulation depending on the AML type?

Lines 222-240, paragraph on “Targeting the MYC/PP2A axis in leukemia”. It is very interesting the review of PP2A activation as an “anti-MYC” therapeutic approach. The section can be improved with an introduction to the “undruggability” of MYC oncoprotein. There are a few reviews about this. Also, the 10058-F4 molecule is presented as a “MYC-MAX inhibitor”  but it is rather an inhibitor of the interaction between MYC and MAX proteins. Other inhibitors of MYC expression as JQ1 and derivatives are being tested in clinical trials (conversely to 10058-F4).

Line 229: Not all but most of the MYC molele is instructred. Actually the DNA binding and MAX interacting region is a basic-helix-loop-helix-leucine zipper domain

Figure 2: the legend mentions that ARPP19 “has been reported to regulate”…  The type of this regulation (i.e., repression) can be indicated, rather than the less informative “regulation”.

Reviewer 2 Report

This is an interesting overview of PP2A in some forms of cancer three of proteins that act as ist endogenous inhibitors and therapeutic options.

However, some alterations and clarifications would improve this impressive opus.

Line 125: change effusively

Line 128: be more specific add that PP2A is a threonine serine phosphatase. The novice might think it could act on tyrosine.

Line 131: give a reference fort the 98 possible isoforms. Which of these do you think are relevant and would be useful to study in more detail?

Line 179: There is an alternative nomenclature for SET namely I2 of PP2A. And Zamuni et al also described a I1 of PP2A. Sometimes if I recall rightly, called PHAP. Please discuss this topic in more detail.

Line 234: clearly there are trials to activate PP2A to stop leukemias. However, please discuss problems: PP2A is also expressed in the heart and the vessel wall. If drugs activate I tone would predict a hypotension via vasodilatation and heart failure due to decreased phosphorylation of Ca Channels in the heart. What is known of these side effects? Is it possible to engineer a blood cell specific activation of PP2A?
